# Germline Aberrations in Pancreatic Cancer: Implications for Clinical Care

**DOI:** 10.3390/cancers14133239

**Published:** 2022-06-30

**Authors:** Raffaella Casolino, Vincenzo Corbo, Philip Beer, Chang-il Hwang, Salvatore Paiella, Valentina Silvestri, Laura Ottini, Andrew V. Biankin

**Affiliations:** 1Wolfson Wohl Cancer Research Centre, Institute of Cancer Sciences, University of Glasgow, Glasgow G61 1QH, UK; pbeer@doctors.org.uk (P.B.); andrew.biankin@glasgow.ac.uk (A.V.B.); 2Beatson West of Scotland Cancer Centre, Glasgow G12 0YN, UK; 3NHS Greater Glasgow and Clyde, Glasgow G4 0SF, UK; 4Department of Diagnostics and Public Health, University of Verona, 37134 Verona, Italy; vincenzo.corbo@univr.it; 5Department of Microbiology and Molecular Genetics, College of Biological Sciences, University of California Davis, Davis, CA 95616, USA; cihwang@ucdavis.edu; 6Comprehensive Cancer Center, University of California Davis, Sacramento, CA 95817, USA; 7General and Pancreatic Surgery Unit, Pancreas Institute, University of Verona Hospital Trust, 37134 Verona, Italy; salvatore.paiella@univr.it; 8Department of Molecular Medicine, Sapienza University of Rome, 00185 Rome, Italy; valentina.silvestri@uniroma1.it (V.S.); laura.ottini@uniroma1.it (L.O.); 9West of Scotland Pancreatic Unit, Glasgow Royal Infirmary, Glasgow G31 2ER, UK; 10South Western Sydney Clinical School, Faculty of Medicine, University of New South Wales, Liverpool, NSW 2170, Australia

**Keywords:** germline, pancreatic cancer, BRCA, PARP inhibitors, precision prevention, familial pancreatic cancer

## Abstract

Pancreatic ductal adenocarcinoma (PDAC) has an extremely poor prognosis and represents a major public health issue, as both its incidence and mortality are expecting to increase steeply over the next years. Effective screening strategies are lacking, and most patients are diagnosed with unresectable disease precluding the only chance of cure. Therapeutic options for advanced disease are limited, and the treatment paradigm is still based on chemotherapy, with a few rare exceptions to targeted therapies. Germline variants in cancer susceptibility genes—particularly those involved in mechanisms of DNA repair—are emerging as promising targets for PDAC treatment and prevention. Hereditary PDAC is part of the spectrum of several syndromic disorders, and germline testing of PDAC patients has relevant implications for broad cancer prevention. Germline aberrations in *BRCA1* and *BRCA2* genes are predictive biomarkers of response to poly(adenosine diphosphate–ribose) polymerase (PARP) inhibitor olaparib and platinum-based chemotherapy in PDAC, while mutations in mismatch repair genes identify patients suitable for immune checkpoint inhibitors. This review provides a timely and comprehensive overview of germline aberrations in PDAC and their implications for clinical care. It also discusses the need for optimal approaches to better select patients for PARP inhibitor therapy, novel therapeutic opportunities under clinical investigation, and preclinical models for cancer susceptibility and drug discovery.

## 1. Introduction

Pancreatic ductal adenocarcinoma (PDAC) is a malignant disease with an extremely poor prognosis [1,2]. Both incidence and mortality continue to rise, and PDAC is predicted to soon become the second leading cause of cancer-related death [3,4]. Major efforts in improving surgical outcomes and progress in therapeutic development have only marginally increased the 5-year overall survival (OS) rate of patients with PDAC over the past 5 decades, and is still less than 10% [3]. Much still needs to be improved to impact the burden of this disease. Currently, most patients (up to 80%) are diagnosed with unresectable disease due to non-specific symptoms and a lack of effective screening strategies. Earlier diagnosis may potentially improve outcomes since surgical resection is the only chance of cure [5]. As a consequence, the identification of biomarkers for early detection is an urgent priority. Patients with advanced tumors are treated with chemotherapy with an unselected approach, either in the neoadjuvant or metastatic setting. The most effective combinatorial regimens, based on phase III clinical trials, i.e., FOLFIRINOX (5-fluorouracil, leucovorin, irinotecan, and oxaliplatin) and nab-paclitaxel plus gemcitabine, only marginally improve the OS of patients, which rarely exceeds one year [6,7,8,9,10,11,12,13]. After progression, less than 50% of patients are eligible for further treatments due to the rapid clinical deterioration typical of this disease. Second-line treatments have a very limited impact on clinical and survival outcomes, and clinical trials remain the optimal therapeutic choice in this setting [14]. 

Multi-omics studies have elucidated the molecular complexity of PDAC, which challenges the development of effective treatments for patients with this tumor [15,16,17,18]. In this context, germline variants in cancer susceptibility genes are emerging as clinically relevant targets for more selective PDAC treatment and prevention. This review summarizes the progress in the field of germline aberrations in PDAC and discusses current challenges, limitations, and implications for clinical care.

## 2. Germline Variants and PDAC Susceptibility

In contrast to somatic mutations, which are acquired during life and arise specifically in tumors, germline variants can be passed from parents to offspring and are associated with hereditary cancer syndromes. Germline pathogenic/likely-pathogenic variants in cancer predisposing genes are prevalent molecular alterations in PDAC (Figure 1) [19,20]. Several studies have shown that 3.8% to 9.7% of patients with PDAC carry a pathogenic germline mutation in genes that predispose them to hereditary cancer syndromes, including familial atypical multiple mole melanoma (*CDKN2A*), Peutz-Jeghers (*STK11*), hereditary breast and ovarian cancer (*BRCA1*, *BRCA2*, *PALB2*, *ATM*), Lynch (*MLH1*, *MSH2*, *MSH6*), and Li-Fraumeni (*TP53*) syndromes [21,22,23,24,25,26,27,28]. In some large single-center datasets, the prevalence of these alterations is as high as 19.8% [29]. Germline deleterious variants in the hereditary pancreatitis genes *PRSS1* and *SPINK1* also confer an increased risk of PDAC [30]. 

Compared with a risk of about 1.5% in the general population, carriers of pathogenic variants in *CDKN2A* and *STK11* have a higher lifetime risk of developing PDAC, which is estimated to be more than 15%. Carriers of pathogenic variants in the breast cancer genes *BRCA2*, *ATM,* and *PALB2* have a moderate lifetime risk, ranging from 5% to 10%, whereas pathogenic variants in *BRCA1* are estimated to confer a lower risk (less than 5%) [31]. More recently, data from a large international consortium of families with hereditary cancer syndromes associated with *BRCA* germline mutations demonstrated that the relative risk (RR) of PDAC was 2.36 (95% CI, 1.51–3.68) for *BRCA1* and 3.34 (95% CI, 2.21–5.06) for *BRCA2*. The absolute risk of PDAC to age 80 was approximately 2.5% (for both *BRCA1* and *BRCA2* carriers) [32]. Pathogenic variants in Lynch syndrome genes and *TP53* are estimated to confer a moderate pancreatic cancer lifetime risk of about 5–10% [33]. 

A family history of PDAC also confers an increased risk. Patients with one or more first-degree relatives affected by PDAC are considered familial pancreatic cancer (FPC) cases [34]. In general, in the presence of a first-degree relative family history, the risk of developing PDAC increases with the number of affected relatives (by up to 4, 6, and 32 times for 1, 2, and 3 or more affected relatives, respectively) [35]. Overall, 80–90% of FPC cases are not attributable to a known genetic cause, suggesting the presence of additional genetic factors involved in PDAC susceptibility that have not yet been identified. 

The recent broad use of cancer predisposing gene panel testing in clinical practice has allowed for the identification of pathogenic variants in a larger number of candidate cancer susceptibility genes, and also in patients without a family history, the so-called sporadic cases, with mutation rates varying among studies [19,36,37,38]. These findings have supported the current National Comprehensive Cancer Network (NCCN) recommendation of performing extended genetic testing on all patients with a diagnosis of PDAC, regardless of family history or age of onset. Genetic testing should be performed with a comprehensive multi-gene panel, including, at a minimum, the genes *ATM*, *BRCA1*, *BRCA2*, *CDKN2A*, *MLH1*, *MSH2*, *MSH6*, *EPCAM*, *PALB2*, *STK11*, and *TP53* [33]. 

The opportunity for extended and universal genetic testing as a standard of care or in the research setting is expanding the probability of identifying clinically actionable germline variants in many genes, but their association with increased risk of PDAC is still uncertain. Indeed, a limited number of proposed candidate susceptibility genes have been consistently associated with an increased risk of PDAC, both in familial and sporadic cases [39,40]. For instance, there is no robust evidence suggesting a significant increased risk of PDAC in mutation carriers of *CHEK2* pathogenic variants [40], although these variants are frequently observed in PDAC patients [19]. Overall, the rarity of pathogenic variants makes it very challenging to define reliable population-based risk estimates, and much larger studies are warranted. Targeted sequencing using comprehensive cancer gene panels may represent the best way to accumulate data to improve the genetic risk assessment for known-candidate genes. On the other hand, a broader genomic approach using whole exome sequencing or whole genome sequencing in selected high-risk families may help define “missing heritability” in PDAC. 

As for most complex diseases, the role of low-penetrance common single nucleotide polymorphisms (SNPs) has been investigated in PDAC using genome-wide association studies (GWAS). According to the GWAS catalogue (accessed May 2021), a total of 200 associations with pancreatic risk were reported; however, few loci reached GWAS statistical significance at a *p*-value threshold of 5 × 10^−8^ and was consistently replicated among many studies (reviewed elsewhere [41]). To date, GWAS-identified loci have been estimated to explain about 4% of the phenotypic variation of PDAC; however, more associated SNPs (up to 1750) are expected to be discovered using larger study populations [29,42]. Since each common variant has a small impact on cancer risk, a polygenic architecture, in which many variants that confer low risk individually act in combination to confer much larger risk in the population, has been suggested as a model of cancer susceptibility. Polygenic risk scores (PRS), developed including GWAS identified loci, and multifactorial risk scores (MRS), developed combining genetic and non-genetic risk factors, were recently shown to improve risk prediction in patients with PDAC [43,44,45]. However, the clinical implementation of these models has yet to be established and deserves further assessment.

## 3. Implications of Germline Variants Identified in PDAC Patients

The identification of germline deleterious variants in cancer susceptibility genes in patients with PDAC or healthy subjects with a significant family history of PDAC (at-risk subjects) has relevant implications for cancer prevention and treatment (Figure 2). 

### 3.1. Preventative Implications

Given that hereditary PDAC is part of the spectrum of several syndromic disorders, germline testing of patients with PDAC has relevant implications for broad cancer prevention. The identification of deleterious variants in well-known cancer susceptibility genes through universal and extended genetic testing, as suggested by recent NCCN guidelines, can potentially become a significant opportunity to reach asymptomatic individuals who are at high risk of certain types of cancers due to genetic predisposition to submit to primary and secondary preventative strategies [46,47]. This can be achieved by applying the strategy of cascade testing, which implies that the direct relatives of mutation carriers are tested in a stepwise manner until all at-risk family members are screened for the specific mutation [48]. As PDAC may share genetic susceptibility with other cancer types (e.g., melanoma, breast, ovarian, colorectal, and prostate cancer), germline testing of healthy relatives through the cascade approach has the potential to indirectly lower overall cancer-related mortality. In this way, PDAC can become a sentinel for the identification of hereditary cancer syndromes previously unknown in the family (Figure 2). Healthy subjects carrying germline variants in *BRCA1*, *BRCA2*, *PALB2*, *CDKN2A*, *ATM,* and MMR genes can be referred to as dedicated intensive surveillance programs, which may include risk-reducing measures for several cancer types (breast, ovarian, prostate cancers, colorectal cancer, and melanoma) as standard of care or in research settings to reduce morbidity and mortality due to those syndromes [49]. 

The management of subjects at risk for PDAC due to familial predispositions is less clear and still debated. First-degree relatives of PDAC patients sharing the same germline deleterious variant are eligible for early detection programs for PDAC. While screening for PDAC is not indicated for the general population due to the relatively low prevalence of this disease, the International Cancer of the Pancreas Screening (CAPS) Consortium [50,51] recommends it to individuals who have a lifetime risk of PDAC > 5% (or a 5-fold increased RR), including those with familial risk only and those diagnosed with predisposing genetic disorders or carrying specific mutations (with or without family history), who may benefit from surgical resection. Familial risk is defined as the presence of (i) at least three affected relatives on the same side of the family, of whom at least one is a first-degree relative; (ii) two affected relatives who are first-degree relatives to each other, of whom at least one is a first-degree relative of the individual to survey; and (iii) at least two affected relatives on the same side of the family, of whom at least one is a first-degree relative [51]. Guidelines of other authoritative international societies, such as those of the American Society of Clinical Oncology [52], NCCN [49], European Society of Medical Oncology [53] and an international panel of experts [54], have also emphasized the importance of PDAC surveillance for high-risk individuals, including those carrying germline pathogenic variants.

Surveillance of these cohorts leads to higher detection rates of PDAC or other pancreatic abnormalities [55]. Although few studies have shown that the surveillance of high-risk subjects could positively impact survival [47,56], it remains uncertain whether PDAC surveillance ultimately reduces PDAC-related mortality. From a clinical standpoint, surveillance should be based on magnetic resonance imaging with cholangiopancreatography, or endoscopic ultrasound, as both have been demonstrated to detect pancreatic disorders at an earlier stage. Regarding the latter, recent reports have shown, unfortunately, that incidental PDAC exists and that 12-month surveillance may sometimes be ineffective [57,58]. In addition, a non-negligible amount of PDAC detected within surveillance is diagnosed at an advanced stage, raising the question of whether this surveillance strategy may be tailored according to an individualized risk-profile [59]. If the screening is negative for pancreatic abnormalities, then it should be repeated annually to possibly improve the early identification of PDAC or pre-malignant lesions. It is not clear at what age pancreatic surveillance should be started. Considering the lifetime risk and anticipation phenomenon that may occur when a germline mutation may be present, 50 years of age (or 10 years earlier than the youngest case of PDAC in the family) is a reasonable cutoff [51,53]. When abnormalities are present, guidelines should be followed for therapeutic management. 

In conclusion, the identification of genetic germline variants in PDAC has relevant implications for cancer risk assessment and broad cancer prevention in family members. PDAC surveillance is challenging, and it should not disregard the genetic background of all individuals at risk of PDAC. Ongoing multicenter studies, such as that of the Pancreatic Cancer Early Detection Consortium (PRECEDE), enrolling thousands of individuals at risk of PDAC [60] aims to assess who should be under surveillance and if surveillance ultimately reduces mortality.

### 3.2. Therapeutic Implications

Emerging data suggest relevant therapeutic implications for PDAC patients with germline pathogenic variants in genes that regulate double-strand break (DSB) repair, such as homologous recombination deficiency (HRD) and MMR (Figure 2). In particular, those in *BRCA1*, *BRCA2*, and increasingly *PALB2* are the most well characterized and include responses to platinum-based chemotherapy agents and poly-ADP (adenosine diphosphate)-ribose polymerase inhibitor (PARPi) [61,62]. Germline alterations in MMR genes associated with the MSI-high phenotype are predictive of the response to immune checkpoint inhibitors (ICIs) [63,64]. Homologous recombination (HR) is the error-free mechanism of DNA repair that repairs DSB, and its functional defects can be exploited to increase the activity of platinum agents or compounds targeting PARP [65,66]. PARP enzymes are involved in the regulation of multiple cellular processes, including the repair of single-strand DNA breaks through base excision repair [67]. PARPi induces cytotoxic effects by inhibiting PARP enzymes, exploiting a synthetic lethal interaction with defects in HR, most of which are due to *BRCA1* or *BRCA2* inactivation [68]. The *BRCA1* and *BRCA2* genes encode critical proteins involved in repairing DSB via HR. The ability to repair double-strand DNA breaks is impaired in cancer cells with deleterious *BRCA* mutations, resulting in an increased reliance on other DNA damage response (DDR) pathways for survival [69]. As a consequence, they become particularly sensitive to the inhibition of the HR and DDR pathways [69,70]. In patients with germline *BRCA* mutations, platinum-based chemotherapy and PARP inhibition increase the chance of cancer cell death [71].

A retrospective study of 71 unresectable PDAC patients with *BRCA1/2* mutations, showed that those treated with platinum agents achieved significantly longer OS than those treated with non-platinum agents (22 vs. 9 months; *p* = 0.039) [72]. Although there is no clear evidence for the superiority of first-line platinum agents over non-platinum agents in PDAC patients with germline *BRCA1/2* or *PALB2* variants, NCCN guidelines recommend FOLFIRINOX or modified FOLFIRINOX or gemcitabine plus cisplatin as first-line chemotherapy for this subgroup of patients [14]. In the phase III POLO study conducted in patients with deleterious or suspected deleterious germline *BRCA* mutation with metastatic PDAC whose disease had not progressed after at least 16 weeks of first-line platinum-based chemotherapy, subsequent maintenance therapy with PARPi olaparib significantly improved progression-free survival (PFS) versus placebo (7.4 vs. 3.8 months, *p* = 0.004) with an objective response rate (ORR) of 23% vs. 12%, respectively [12]. These results led to the approval of olaparib in multiple countries as a maintenance therapy after platinum-based first-line treatment in patients with advanced PDAC associated with germline *BRCA* mutation. Retrospective studies suggest that preoperative platinum-based chemotherapy is most effective in patients with germline *BRCA* mutation [73,74] highlighting the importance of having *BRCA* status available at the time of diagnosis, even in patients with early stage disease [75]. In addition, a phase II trial is ongoing to investigate the addition of olaparib following completion of surgery and chemotherapy in patients with resected PDAC and pathogenic mutations in *BRCA1*, *BRCA2*, or *PALB2* (The APOLLO Trial, ClinicalTrials.gov Identifier: NCT04858334). 

Another PARPi, rucaparib, was tested in a phase II study in patients with a pathogenic germline or somatic variant in *BRCA1*, *BRCA2*, or *PALB2*. The results were promising, with a 37% response rate in patients with somatic or germline variants in these genes [76]. PARPi has also been investigated as a monotherapy or in combination with chemotherapy. In phase II trials of patients with advanced PDAC and germline *BRCA1* or *BRCA2* mutations, olaparib and rucaparib were associated with 21.1% and 21.7% ORR [77,78]. 

Veliparib was tested in association with gemcitabine and cisplatin in a phase I trial of advanced untreated PDAC patients with *BRCA1*, *BRCA2*, and *PALB2* mutations, with no significant differences in terms of OS, PFS, or ORR in patients in the investigational arm (treated with veliparib) [79]. A randomized phase II trial of this regimen is currently ongoing (ClinicalTrials.gov Identifier: NCT01585805). 

Germline mutations in MMR genes account for around 1% of PDAC cases [80]. MMR increases the fidelity of DNA replication by dealing with the misincorporation of nucleotides [81,82] and relies on highly conserved proteins encoded by the mutS and mutL homologue genes, such as *MSH2* and *MLH1* [82]. When MMR is defective (dMMR) due to genetic or epigenetic inactivation of MMR genes, the inability to correct DNA replication errors leads to hypermutated genomes with a peculiar mutation pattern affecting dinucleotide repeats, i.e., MSI. dMMR and MSI are observed in sporadic tumors due to double somatic inactivation and also in the context of Lynch syndrome, which is caused by germline mutation of one of the MMR genes (*MLH1*, *MSH2*, *MSH6,* or *PMS2*) or epigenetic silencing of *MSH2* consequent to germline mutation in EPCAM [82]. Although dMMR/MSI is rare in PDAC, it identifies a distinct subgroup of patients with unique clinical, pathological, and genomic features. dMMR PDAC are enriched for several markers of immune activation (including high tumor mutational burden and neoantigen load, chemokine signatures, and cytolytic activity), are less likely to have mutations in usual PDAC driver genes like *KRAS* and *SMAD4*, but more likely to have mutations in genes that drive cancers with microsatellite instability like *ACV2RA* and *JAK1* [83].

Anti-PD-1 therapy pembrolizumab showed antitumor activity in dMMR/MSI malignancies regardless of histotype and is now approved for the treatment of patients with unresectable or metastatic dMMR/MSI-high solid tumors, including PDAC, after prior conventional chemotherapies [64]. In a small phase II clinical trial, pembrolizumab achieved an 18% ORR in advanced MSI-high PDAC [63]. 

There is also basic and translational evidence indicating the synergistic effects of PARPi and ICIs. PARPi-mediated unrepaired DNA breaks modulate the tumor microenvironment by several molecular and cellular mechanisms that might induce a response to ICIs. These include increased genomic instability, activation of immune pathways, and induced PD-L1 expression on cancer cells [84]. Following promising results from breast and ovarian cancer, PARPi are being investigated in combination with ICIs in PDAC patients with HRD, including those with germline mutations (ClinicalTrials.gov Identifier: NCT04666740, NCT04409002).

In conclusion, pathogenic variants in several cancer susceptibility genes are therapeutically actionable with platinum, PARPi, and ICI therapy. Other potential therapeutic targets need to be identified and tested in the clinic to improve the outcomes of patients carrying germline pathogenic variants. There is substantial activity in exploring novel DDR agents and combinations of agents to target these mechanisms in many cancer types. Most notably, the potential ability to generate “synthetic” synthetic lethality, where a drug induces a defect in DDR, can be exploited through the same mechanisms as inherited defects in DDR. Whether non-BRCA mutations in HR genes confer PARPi sensitivity needs to be addressed in the future, although the rarity of the non-BRCA pathogenic variants makes it difficult to evaluate the clinical benefit for those patients. 

A selection of interventional clinical trials currently recruiting PDAC patients with germline mutations is reported in Table 1. 

## 4. Challenges in Targeting BRCA Mutations in PDAC

Despite encouraging results, treatment with PARPi in PDAC patients with germline *BRCA* mutations is challenging due to the primary and secondary therapeutic resistance that inevitably occurs. Inhibition of PARP activity generates DNA lesions, such as collapsed replication forks, which are repaired by HR. Therefore, it is thought that BRCA1/2 deficient or HR-deficient cancer cells are selectively sensitive to PARPi because the collapsed replication forks are not properly repaired in HR-deficient cancer cells. However, the underlying molecular mechanism of how PARPi induces anti-cancer effects has not been fully understood. In addition, the consequence of BRCA1/2 deficiency appears to have a lineage-dependent effect, and it is likely to be seen in the context of other cancer susceptibility gene mutations [85]. A recent study of the lineage dependency of BRCA-mediated phenotypes highlights the possibility that a distinct epigenetic landscape from a different lineage of cancer cells may influence therapeutic responses to PARPi. Likewise, many epigenetic drugs, such as bromodomain inhibitors, HDAC inhibitors, and DNMT inhibitors, sensitize cancer cells to PARPi, likely converting HR-proficient cancer cells to HR-deficient cells, termed ‘induced BRCAness’ [86,87,88,89,90]. Similarly, genetic alterations in epigenetic regulators, such as ARID1A deficiency and ETS fusion, result in increased sensitivity to PARPi in breast and prostate cancer [91,92]. It is also important to note that several epigenetic regulators are frequently mutated in PDAC, and it is therefore highly possible that epigenetic regulators, transcription regulation, and DDR pathways converge in the early stage of PDAC progression and may play a role in PARPi resistance. 

Although several mechanisms of resistance to PARPi have been described in breast and ovarian cancer patients, data on PDAC are still limited. The landscape of somatic mutations in BRCA1/2-associated PDAC is essentially indistinguishable from that of sporadic cancers. As it stands, the two types of disease might share the same evolutionary path, including an estimated timeline of two decades from inception to metastatic disease [93,94,95]. Although speculative, a possible implication would be that HR defects are indispensable founding events for tumor maintenance in a subset of germline *BRCA1/2* mutation carriers. If these represent possible explanations of primary resistance, the clinical response profile of PDAC patients to PARPi olaparib also suggests the rapid emergence of secondary resistance [12]. Given that patients received platinum prior to PARPi, it would be difficult to gauge the specific contribution of PARPi to secondary resistance in this context. Unfortunately, the confounding factor of prior therapy with platinum is constant in many clinical trials with PARPi in PDAC, as well as in other *BRCA1/2*-associated cancer types. 

HR deficiency is considered a prerequisite for response to PARP inhibition. Tumors that have acquired resistance to PARP inhibition can be either HR deficient or HR proficient (Figure 3). In HR-proficient tumors, the genetic mutation in the HR gene that results in the HR-deficient phenotype is repaired by a reversion mutation. Secondary mutations either remove the original pathogenic mutation or result in restoration of the open reading frame [96,97,98]. Most of these mutations conferring therapeutic resistance are deletions surrounded by sequence microhomologies, which can be explained by end-joining repair mechanisms [99,100]. Mechanisms of resistance to PARP inhibition where the tumor is still HR deficient are less well understood. Preclinical studies, including forward genetic screens and patient xenograft models, have identified several potential pathways, including loss of activity of the Shieldin complex, TP53BP1 or SLFN11, mutations in PARP1, resulting in loss of PARPi binding, or enhanced drug efflux, resulting in loss of efficacy [101,102,103]. At present, however, robust data corroborating these mechanisms in patients have not been reported. It has been demonstrated that most tumors that develop resistance to PARPi have a hyper-activated ATR/CHK1 pathway. PARP inhibition in *BRCA*-mutated cancer cells increases reliance on the ATR/CHK1 pathway for genome stability [104]. This provides the rationale for simultaneous inhibition of PARP and ATR/CHK1 pathway. A phase II trial (ClinicalTrials.gov identifier: NCT03462342) is currently investigating olaparib in association with a novel ART inhibitor, AZD6738. 

In conclusion, treatment with PARPi in PDAC patients with *BRCA* mutations presents many challenges. Optimal approaches to selecting patients with PDAC for PARPi therapy have yet to be described. Sensitivity to platinum chemotherapy is expected to enrich PARPi sensitivity, although platinum chemotherapy elicits cell death through both HR-dependent and non-HR-dependent mechanisms, thus reducing predictive power. The presence of an inherited mutation in *BRCA1/2* or another HR pathway gene may also correlate sub-optimally to PARPi response due to incomplete penetrance. Alternative approaches are based on the detection of the phenotypic consequences of HR deficiency, either at the genomic or functional levels. The genomic scar of HR deficiency is highly characteristic and is likely to represent a strong biomarker of PARPi sensitivity in treatment naïve patients [105]. At present, however, this information can only be acquired through sequencing of the entire, or a large proportion of the cancer genome, which is currently not feasible in routine clinical care for most patients. Surrogate biomarkers for HR deficiency have been reported, such as enumeration of genomic deletions using loss of heterozygosity; however, the sensitivity, specificity, and inter-assay reproducibility of this approach has yet to be determined and is currently the focus of an international harmonization project (https://friendsofcancerresearch.org/hrd, accessed on 8 June 2022). Importantly, genomic scar assays will not distinguish between the two main pathways to PARPi resistance (shown in Figure 3), as tumors that have undergone secondary mutations to reactivate the HR pathway will still carry the genomic scars of HR deficiency. An alternative approach is to deploy a functional assay of HR competence, such as the detection of RAD51 foci in tumor samples [106]. This approach has the potential to select patients for upfront PARPi therapy and to distinguish the key pathways to PARPi resistance, which may be of relevance in informing future lines of therapy.

## 5. Preclinical and Translational Research in Hereditary PDAC 

Preclinical and translational research is essential to improve our understanding of PDAC susceptibility and to facilitate the development of therapeutic strategies for patients with germline mutations. While approximately 10% of PDAC patients may harbor germline mutations, only a few pre-clinical models are commercially available in this field. CAPAN1 is the most commonly used BRCA2-deficient PDAC cell line, harboring *BRCA2* c.6774delT truncating mutation [107]. In addition, PL11 and Hs766T harbor genetic alterations in the Fanconi anemia pathway genes *FANCC* (null mutation) and *FANCG* (nonsense mutation), respectively [108]. To our knowledge, there are no detailed genomic annotations associated with commercially available PDAC cell lines or other preclinical models, and this hampers the preclinical usage of PDAC cell lines in the context of FPC. Genetically engineered mouse models have been critical for basic PDAC research and the preclinical evaluation of therapeutic strategies. One of the representative PDAC GEMM is the KPC mouse model, which harbors oncogenic *KRAS* G12D and a gain-of-function *p53* R172H or R270H mutation (equivalent to human *TP53* R175H or R273H) or *p53* null mutation specifically in pancreatic epithelial cells driven by a Pdx1-Cre or Ptf1-Cre transgenic allele [109,110]. Since *KRAS* and *TP53* mutations are the most common genetic alterations in PDAC, putative tumor suppressors or oncogenes have been modeled in the Kras mutant background or *Kras/Trp53* double mutant background (reviewed by Guerra and Barbacid [111]). Although many genes are associated with hereditary PDAC, a few genes have been experimentally shown to contribute to PDAC progression in vivo using GEMM. Among cancer predisposing genes, mutations in *BRCA2* and *ATM* play a role in PDAC progression [112,113], whilst the role of other genes remains to be defined. One of the challenges in modeling hereditary PDAC with GEMM is that the generation of conditional knockout alleles for individual cancer predisposing genes is time-consuming. In addition, PDAC mouse models need to be crossed with multiple oncogenic alleles, such as *Kras* mutations and pancreas-specific Cre alleles. Another critical issue related to hereditary PDAC is that genetic mutations are introduced at the embryonic development stage, which complicates the question of whether cancer predisposing genes play a role in the inception of key driver mutations, such as oncogenic mutations in *KRAS* in other tumor suppressors. Available GEMMs only allow us to address whether these mutations in cancer predisposing genes can cooperate with oncogenic *Kras* mutations or other driver mutations for PDAC progression.

In conclusion, various preclinical models of human PDAC are available for basic and translational research, including PDAC cell lines, patient-derived xenografts (PDXs), and patient-derived organoids (PDOs) [114]. Each preclinical model has its own advantages and disadvantages. Thus, the optimal model for each study should be determined based on specific scientific and clinical questions. To evaluate the efficacy of PARPi and other therapies for PDAC patients, a lack of detailed genomic annotations in the currently existing preclinical models is an issue that needs to be addressed. The use of isogenic cell lines or other preclinical models with CRISPR or other genetic engineering approaches is ideal for addressing a mutation-specific drug response in hereditary PDAC. In this way, the possibility of confounding effects that may come from other genetic mutations or other backgrounds can be excluded. In addition, the use of preneoplastic cells (e.g., PanIN-derived organoids) could be useful to address the effect of cancer predisposing gene mutations in the early stage of PDAC progression without generating new GEMM for hereditary PDAC [115]. 

## 6. Conclusions

Germline pathogenic variants are prevalent and clinically relevant in PDAC. The American Society of Clinical Oncology and the NCCN guidelines recommend risk assessment and extended germline testing for all individuals with PDAC irrespective of personal or family history of cancer, age, or ethnicity to maximize the opportunity for targeted therapeutic interventions for patients and cancer prevention in the families. The proposed algorithm for germline genetic testing in PDAC and high-risk subjects is shown in Figure 4. Treatment implications include the use of ICI for MMR-d PDAC and PARPi therapy with olaparib as a maintenance strategy in platinum-sensitive *BRCA* mutation carriers. Whether mutations in other non-BRCA genes can be successfully targeted with PARPi therapies remains to be addressed in both preclinical and clinical settings. Continued research on the actionability of germline variants in PDAC, on the molecular mechanism of how PARPi exerts anti-cancer effects, and on how germline variants contribute to PDAC progression is essential to improve our understanding of PDAC and to facilitate the development of therapeutic and preventative strategies for patients and their families. 

## Figures and Tables

**Figure 1 cancers-14-03239-f001:**
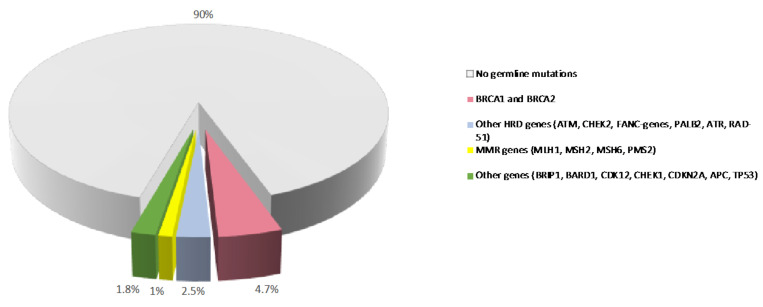
Prevalence of germline variants in PDAC. Prevalence of germline mutations in PDAC patients from published studies. MMR: mismatch repair. HR: homologous recombination.

**Figure 2 cancers-14-03239-f002:**
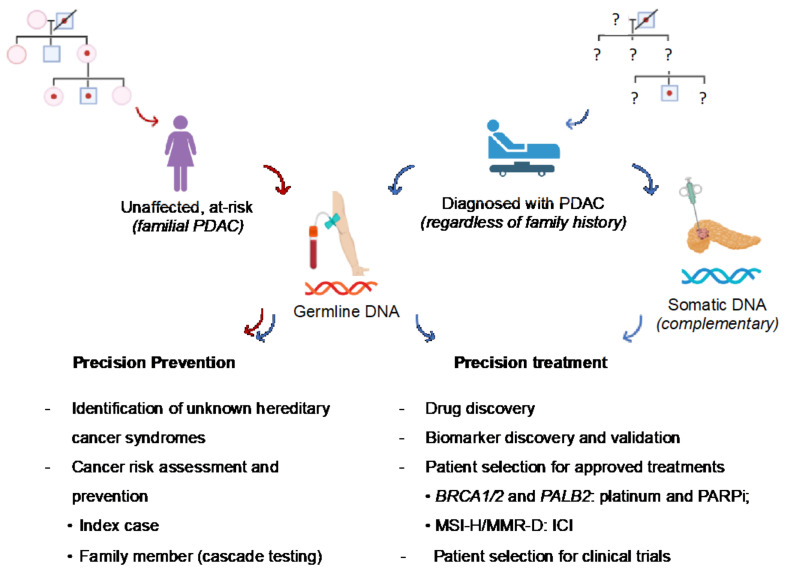
Overview of the clinical implications of germline variants in PDAC patients and at-risk subjects. Pathogenic germline variants are of key interest in PDAC owing to their therapeutic actionability and implications for cancer prevention (downstream identification of at-risk relatives and possible hereditary cancer syndromes previously unknown in the family). Red arrow: pathway of healthy individuals at risk of PDAC based on family history. Blue arrow: pathway of patients with PDAC who should be tested for germline variants at diagnosis, regardless of family history. Complementary somatic analysis of tumor tissue may help the therapeutic decision (not standard recommendation, only in the research context). MMR-D: mismatch repair-deficiency; MSI-H: microsatellite instability–high; PARPi: poly-ADP (adenosine diphosphate)-ribose polymerase inhibitors; ICI: immune checkpoint inhibitors.

**Figure 3 cancers-14-03239-f003:**
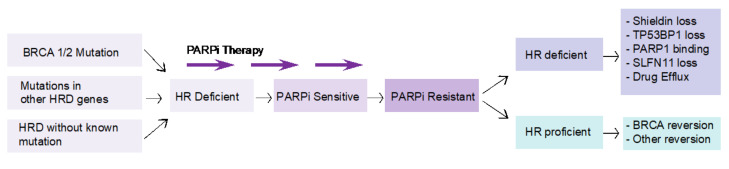
Mechanisms of resistance to PARPi according to HR status. Homologous recombination (HR) deficiency is considered a prerequisite for response to PARP inhibition. Tumors that have acquired resistance to PARP inhibition can be either HR deficient or HR proficient. In HR-proficient tumors, the genetic mutation in the HR gene that results in the HR-deficient phenotype is repaired by a reversion mutation. Mechanisms of resistance to PARP inhibition where the tumor is still HR deficient include loss of activity of the Shieldin complex, TP53BP1 or SLFN11, mutations in PARP1, or enhanced drug efflux.

**Figure 4 cancers-14-03239-f004:**
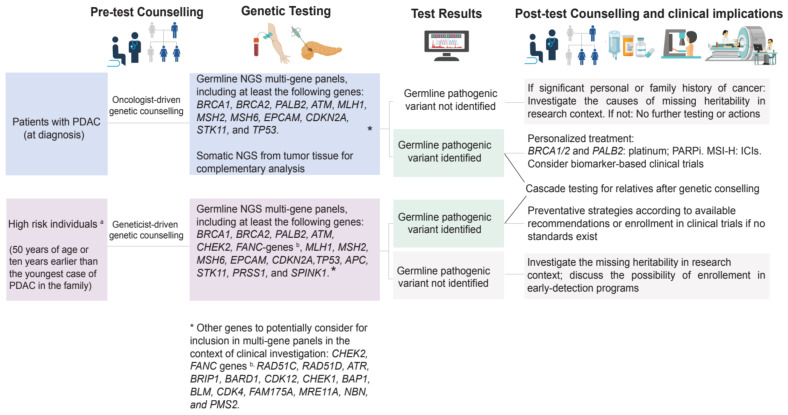
Suggested algorithm for germline testing for healthy individual at high risk of PDAC and individuals diagnosed with PDAC. MSI-H: microsatellite instability–high; PARPi: poly-ADP (adenosine diphosphate)-ribose polymerase inhibitors; ICIs: immune checkpoint inhibitors.

**Table 1 cancers-14-03239-t001:** Selection of interventional clinical trials for PDAC patients with germline mutations is currently ongoing.

NCT Number	Title	Condition(s)	Interventions	Phase	Start Date
NCT04493060	Niraparib and Dostarlimab for the Treatment of Germline or Somatic BRCA1/2 and PALB2 Mutated Metastatic Pancreatic Cancer	Metastatic Pancreatic cancer	Drug: Niraparib |Biological: Dostarlimab	2	December 2020
NCT04548752	Testing the Addition of Pembrolizumab, an Immunotherapy Cancer Drug to Olaparib Alone as Therapy for Patients With Pancreatic Cancer That Has Spread With Inherited BRCA Mutations	Metastatic Pancreatic cancer	Drug: Olaparib|Biological: Pembrolizumab	2	December 2020
NCT03553004	Niraparib in Metastatic Pancreatic Cancer After Previous Chemotherapy (NIRA-PANC): a Phase 2 Trial	Metastatic Pancreatic Cancer	Drug: Niraparib	2	January 2019
NCT04858334	A Randomized Study of Olaparib or Placebo in Patients With Surgically Removed Pancreatic Cancer Who Have a BRCA1, BRCA2 or PALB2 Mutation, The APOLLO Trial	Resected Pancreatic Cancer (Adjuvant setting)	Drug: Olaparib|Drug: Placebo Administration	2	April 2021
NCT04890613	Study of CX-5461 in Patients With Solid Tumours and BRCA1/2, PALB2 or Homologous Recombination Deficiency (HRD) Mutation	Advanced Solid Tumor	Drug: CX-5461	1	September 2021
NCT04171700	A Study to Evaluate Rucaparib in Patients With Solid Tumors and With Deleterious Mutations in HRR Genes(LODESTAR trial)	Advanced Solid Tumor	Drug: Rucaparib	2	November 2019
NCT04673448	Niraparib and TSR-042 for the Treatment of BRCA-Mutated Unresectable or Metastatic Breast, Pancreas, Ovary, Fallopian Tube, or Primary Peritoneal Cancer	Advanced Unresectable or Metastatic Breast, Pancreas, Ovary, Fallopian Tube, or Primary Peritoneal Cancer	Biological: Dostarlimab|Drug: Niraparib	1	November 2021
NCT04300114	A Study of Maintenance Treatment With Fluzoparib in gBRCA/PALB2 Mutated Pancreatic Cancer Whose Disease Has Not Progressed on First Line Platinum-Based Chemotherapy	Metastatic Pancreatic Cancer	Drug: Fluzoparib|Drug: Placebo	3	August 2020
NCT04150042	A Study of Melphalan, BCNU, Vitamin B12b, Vitamin C, and Stem Cell Infusion in People With Advanced Pancreatic Cancer and BRCA Mutations	Metastatic Pancreatic Cancer	Drug: Melphalan|Drug: BCNU|Drug: Vitamin B12B|Drug: Vitamin C|Drug: Ethanol|Device: Autologous Hematopoietic Stem Cells	1	January 2021

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
