# Peer review of "Germline Aberrations in Pancreatic Cancer: Implications for Clinical Care"

_cancers, 2022, doi:10.3390/cancers14133239_

Round 1

Reviewer 1 Report

The review article entitled ‘Germline aberrations in Pancreatic Cancer: implications for clinical care’ is very interesting and it describes germline aberrations in PDAC and their implications for clinical care. The authors have nicely summarized the latest developments in the field. However, The article can be improved. Below are some comments.

1. The review article describes the various genes such as ATM, BRCA1, 101 BRCA2, CDKN2A, MLH1, MSH2, MSH6, EPCAM, PALB2, STK11, TP53 that are implicated in PDAC, however, the authors need to describe if they can be used as a biomarker for PDAC disease progression.

2. Yes, specific targeting of these genes by CRISPR technology by HDR or NHEJ repair mechanisms will open up more avenues for the treatment of PDAC, the authors could elaborate about it as the recent evolvement of invivo editing and base editing to target  specific genes would be interesting. It would be interesting to know if there are any specific mutations in the above mentioned genes that could specifically be targettted for therapeutic purposes. A table describing them would be more illustrative.

3. The authors could elaborate more the future therapeutic interventions that could be targeted similar to PARP inhibitors.

4. There is emerging evidence related to targeting of EMT genes such as Snail, slug and other genes, the authors could include these in the discussion.

Author Response

  1. The review article describes the various genes such as ATM, BRCA1, 101 BRCA2, CDKN2A, MLH1, MSH2, MSH6, EPCAM, PALB2, STK11, TP53 that are implicated in PDAC, however, the authors need to describe if they can be used as a biomarker for PDAC disease progression.

We thank the reviewer for this suggestion, and the detection of circulating DNA to gauge treatment response and disease progression is very topical. Unfortunately, at this stage there are little data as to whether these markers have utility for the detection of disease progression for somatic mutations, and no data that we are aware of that utilize germ line mutations.

  1. Yes, specific targeting of these genes by CRISPR technology by HDR or NHEJ repair mechanisms will open up more avenues for the treatment of PDAC, the authors could elaborate about it as the recent evolvement of in vivo editing and base editing to target specific genes would be interesting. It would be interesting to know if there are any specific mutations in the above mentioned genes that could specifically be targeted for therapeutic purposes. A table describing them would be more illustrative.

We thank the reviewer for this insight. The identification of synthetic lethal strategies using the abovementioned approaches is very much in the early discovery stages apart from BRCA itself and data are scant.

  1. The authors could elaborate more the future therapeutic interventions that could be targeted similar to PARP inhibitors.

As mentioned above, these strategies are still in early discovery and not yet in the clinic, however there are many combinations being tested preclinically and some in early clinical trials, but none in pancreatic cancer. The following sentence addressing this question has been added:

There is substantial activity in exploring novel DDR agents and combinations of agents to target these mechanisms in many cancer types. Most notably the potential ability to generate “synthetic” synthetic lethality, where a drug induces a defect in DDR, which can be exploited through the same mechanisms as inherited defects in DDR

  1. There is emerging evidence related to targeting of EMT genes such as Snail, slug and other genes, the authors could include these in the discussion.

We thank the reviewer for this comment, but we are not aware of genes in these pathways where germline mutations predispose to, or drive pancreatic cancer.

Reviewer 2 Report

Dear Authors,

I find your review article to be thoroughly written. Pancreatic cancer is a topic of interest, and this review approach does cover it well.

All in all, I am satisfied with the quality of the content and its presentation in this manuscript, recommending it for publication with minor corrections to be addressed:

-       In text referencing should be formatted according to the MDPI style

-       So should the ones in the References section

-       Minor English language mistakes should be addressed in a thorough reading by the authors

-       Section numbering should be used and formatted according to the MDPI style

-       The manuscript lacks the sections pertaining to Authors contributions, conflict of interest, etc.

Final remark, is it necessary that 10 out of 116 references are citing work that the authors have previously co-authored? Please explain the usage of the following original work in the context of this publication:

-       [17,19,62,76] Casolino

-       [19,116] Corbo

-       [19,54] Beer

-       [115,116] Hwang

-       [19,55,76] Paiella

-       [32] Silvestri 

-       [16,17,62] Biankin 

Were these the only groups working on the original articles revealing the most relevant results in the field? If yes, then referencing is appropriate. If the studies are reviews, please try referencing original work whenever possible, and remove the potentially inappropriate references. 

Best regards,

Author Response

All the points below have been addressed except "Section numbering should be used and formatted according to the MDPI style" which is not applicable in our care.

-    In text referencing should be formatted according to the MDPI style

-       So should the ones in the References section

-       Minor English language mistakes should be addressed in a thorough reading by the authors

-       Section numbering should be used and formatted according to the MDPI style

-       The manuscript lacks the sections pertaining to Authors contributions, conflict of interest, etc.

Final remark, is it necessary that 10 out of 116 references are citing work that the authors have previously co-authored? Please explain the usage of the following original work in the context of this publication:

-       [17,19,62,76] Casolino

17 is the most comprehensive recent review on molecular subtypes of PDAC

19 is the only systematic review and Meta-Analysis on HRD and germline mutations in PDAC which have analysed all the literature on this topic and supports universal and germline testing in PDAC

62 is a real-time report of clinical implications of germline mutations in genes other than BRAC in PDAC, published in The Lancet Oncology. It reports new findings about predictive value of VUS in PALB2

76 is the only available paper on the role of molecular profiling and germline mutations in preoperative management of PDAC.

-       [19,116] Corbo

19 is the only systematic review and Meta-Analysis on HRD and germline mutations in PDAC which have analysed all the literature on this topic and supports universal and germline testing in PDAC

116 One of the most comprehensive paper on organoid models of human and mouse ductal pancreatic cancer published in Cell

-       [19,54] Beer

19 is the only systematic review and Meta-Analysis on HRD and germline mutations in PDAC which have analysed all the literature on this topic and supports universal and germline testing in PDAC

54: Clinical practice guidelines for BRCA1 and BRCA2 genetic testing from international experts, which includes relevant information about BRCA testing in PDAC

-       [115,116] Hwang

115 One of the most comprehensive paper on preclinical models of pancreatic cancer published so far

116 One of the most comprehensive paper on organoid models of human and mouse ductal pancreatic cancer published in Cell

-   [19,55,76] Paiella

19 is the only systematic review and Meta-Analysis on HRD and germline mutations in PDAC which have analysed all the literature on this topic and supports universal and germline testing in PDAC

55 Extremely relevant paper on screening/surveillance programs for pancreatic cancer in familial high-risk individuals (it is a systematic review and proportion meta-analysis of screening results)

76 is the only available paper on the role of molecular profiling and germline mutations in preoperative management of PDAC.

-       [32] Silvestri it is the most relevant paper published so far about this topic, published in JCO

-       [16,17,62] Biankin

16 and 62 are extremely relevant papers in the topic molecular aberrations in PDAC (16) and clinical implications of germline mutations in PDAC (62). 62 is a real-time report of clinical implications of germline mutations in genes other than BRAC in PDAC, published in The Lancet Oncology. It reports new findings about predictive value of VUS in PALB2

17 is the most comprehensive recent review on molecular subtypes of PDAC

Were these the only groups working on the original articles revealing the most relevant results in the field?

Yes